# Gutka consumption and dietary partialities explaining anemia in women of a coastal slum of Karachi, Pakistan: A mixed-method study

Ameer Muhammad[1]*, Sarah Saleem[2], Daniyaal Ahmad[3‡], Eleze Tariq[3‡], Yasir Shafiq[4]

1 VITAL Pakistan Trust, Karachi, Pakistan, 2 Department of Community Health Science, The Aga Khan University, Karachi, Pakistan, 3 Medical College, The Aga Khan University, Karachi, Pakistan, 4 Department of Pediatrics and Child Health, The Aga Khan University, Karachi, Pakistan

☺ These authors contributed equally to this work.
‡ DA and ET also contributed equally to this work.
* ameer.muhammad@vitalpakistantrust.org

**Data Availability Statement:** Data is available and attached as supplementary file.

**Funding:** This study was not funded by any external sponsor. However, the author received

## Abstract

### Background

Limited literature is available on the dietary pattern and its consequences on health of women living in coastal slums of Karachi, Pakistan.

### Material and methods

The study employed a mixed-method approach where concurrent quantitative and qualitative assessments were carried out. An analytical cross-sectional survey was conducted to collect information on demographic, household, obstetrics characteristics, and dietary pattern of married women of reproductive age (MWRA). Blood samples were collected to identify the hemoglobin level to determine anemia. For the qualitative component, focus group discussions were carried out with women and in-depth interviews with shopkeepers to understand the availability of food items at household level and in local markets respectively. In addition, observational visits were carried out at different points in time to the local market to document the availability of iron-rich foods for the community.

### Results

The overall prevalence of anemia in sample population was 68.0%. Women with no formal education (AOR: 2.93 95% CI: 1.90–4.52), who consumed gutka (AOR: 2.84 95% CI: 1.81–4.46), did not eat red meat (AOR: 1.68 95% CI: 1.06–2.65), and only had seafood (AOR: 4.56 95% CI: 1.38–15.02) were more likely to be anemic as compared to their counterparts. Qualitative data revealed that any kind of meat and fruits were beyond the reach of community people due to non-affordable cost. A high percentage of women used a locally produced recreational substance known as gutka which gives them a feeling of wellbeing and suppresses hunger.

support from the VITAL Pakistan Trust through existing maternal, neonatal and child health programs to conduct laboratory assessments. VITAL Pakistan Trust had no role in study design, data collection and analysis, decision to publish, or preparation of the manuscript.

**Competing interests:** The authors have declared that no competing interests exist.

**Abbreviations:** AOR, Adjusted odds ratios; COR, Crude odds ratios; FGDs, Focus Group Discussions; HFIAS, Household Food Insecurity Access Scale; IDIs, In-depth interviews; MWRA, Married women of reproductive age; RBCs, Red Blood Cells.

## Conclusion

In our study population, lack of access to diversity of food items, illiteracy, and use of gutka are the statistically significant factors which are associated with anemia in married women of this coastal slum area. The lack of demand for diversity in food is related to poverty and preference of spending money on gutka.

## Introduction

Anemia is a blood disorder which is related to decrease in number of red blood cells (RBCs) and reduction in oxygen-carrying capacity of RBCs to meet physiologic needs of the body organs [1]. It is one of the major global public health threats imposing risk to millions of lives in both developed as well as developing world (1). Anemia can affect all age groups, but women in reproductive age, young children and adolescent are effected most [2]. The common causes of anemia in married women of reproductive age (MWRA) between of 18–49 years of age include: iron deficiency, other nutrition causes, blood loss as a result of menstruation, parasite infestation, acute and chronic infections and illnesses like malaria, cancer, tuberculosis, and HIV that can result in lower hemoglobin levels [1, 2]. Further, presence of other micronutrient deficiencies, like vitamins A and B12, folate, riboflavin, and copper deficiency can enhance the risk of developing anemia [1]. Specific to nutrition related causes, the women in low- and middle- income countries are prone to develop anemia due to factors such as type of food consumed, unbalanced diet, nutritional status of the woman before conception and food insecurity at the household [3].

The global estimates indicate that around 468.4 million (30.2%) non-pregnant MWRA, are anemic [4]. Sub Saharan Africa and South Asia regions are the most affected regions with high prevalence, of anemia i.e., 47.5% and 35.7% respectively [5]. Similarly, the situation in the Eastern Mediterranean region has alarmingly high prevalence of anemia i.e. around 32.4% [5]. The health risks double if anemic women conceive leading to adverse maternal and newborn health outcomes [6]. Around 20% of maternal deaths are related to preexisting maternal anemia and stunting in women; adding 115,000 deaths per annum to the total burden of maternal deaths due to obstetric complications [7].

The maternal health indicators like anemia and other are grossly linked poverty and food insecurity [8]. Rapid urbanization has affected already poor populations most, creating huge impact of health of the vulnerable [9]. These are the people who live in slums and in coastal regions [9, 10]. Currently one in three urban residents live in slums [11]. By 2050, nearly 2.5 billion of population globally is expected to be absorbed by urban areas in developing countries [11]. Populations living in these urban slums, which are very close to or within metropolitan cities, are impacting overall health indicators of urban wealthy and well-off counterpart [12–16]. The poorest of the poor and most vulnerable are the women and children of these areas who are 50% more at risk of death compared to the richer urban counterparts and are in more devastating state of health as compared to rural set-up [12–16].

The anemia prevalence among MWRA living in slums is high, especially in the coastal belts [17]. These slums are grossly disconnected from the rest of the urban population, and poverty in these areas is linked to poor health, food insecurity, hunger, and poor dietary consumption [18]. Monotonous diet consumption, such as seafood, is also common in these areas, and accessibility to diverse food is limited because of seasonality of earnings related to fishing [19]. Further, poverty may lead to high substance abuse in the form of smokeless tobacco like

"gutka" and others [20, 21]. The use of gutka is emerging as the most common substance abuse in the form of smokeless tobacco, which contain betel nut, catechu, slake lime, flavorings and other agents [22, 23]. There are a number of ingredients that inhibit iron absorption and storage in the human body, therefore increasing the risk of iron depletion and anemia [24–29].

There is a high level of undernutrition among MWRA in Pakistan, the world's sixth most populous country. The results of a recent nutritional survey indicate that 43.0% of women of reproductive age who are not pregnant are anemic in Pakistan [30]. The urban and rural estimates are reported as 40.6% and 44.6%, respectively [30]. The prevalence of anemia in Sindh province is estimated at 45.7%, and in Karachi, the largest city, it is estimated at 35% [30]. There is no data on anemia in districts specific to coastal slums which is expected to be high due to poor socioeconomic status and patterns of behavior there. As far as dietary patterns are concerned, only 27.6% of the population in Pakistan consumes a diet that is minimally diverse on a daily basis [30]. In areas hosting coastal slums, the minimal diversity in the daily diet is below 20.3% [30]. The consumption of meat from different sources in these districts remains between 55.45% and 58.4% [30]. However, specific indicators for coastal slums are not reported [30] Additionally, gutka is eaten in many South Asian countries, including Pakistan. According to a study conducted in Thatta, a rural coastal belt in Sindh, 71.4% and 75.4% of these women were anemic [31]. However, information on anemia prevalence among MWRA in coastal slums of Karachi, their dietary patterns and use of substance is missing from the national surveys and other research mandate.

Research on gutka consumption and maternal health is scarce, especially in resource-poor settings. A few studies have suggested that smoking smokeless tobacco, such as gutka, results in maternal anemia, stillbirth, prematurity, and a low birth weight in the infant [32, 33]. Therefore, this study aimed to assess the magnitude of anemia among non-pregnant MWRA at one of the oldest coastal slums of Karachi and determine the impact of gutka consumption and dietary patterns among these MWRA on anemia burden. Furthermore, deep dived to gather the community insights on the use of gutka and accessibility to iron-rich diet was assessed. The purpose of this paper is to report the findings and learnings of the study conducted at Rehri Goth. From this, some key recommendations are suggested for improving maternal health there.

## Material and methods

### Study design

The study employed a 'mixed-methods approach' by using a 'quantitative' as well as 'qualitative' approaches. An analytical cross-sectional survey was conducted to assess the prevalence of anemia and its predictors. Further, in-depth interviews (IDIs), focus group discussions (FGDs) as well as market survey were conducted for qualitative approach. An exploratory design was chosen gain insights into food availability, access, and diversity. Information on gutka use was also explored.

### Study setting

The study was conducted in Rehri Goth, one of the oldest coastal slums located in Bin Qasim town, Karachi. The total population of the area is approximately 42,000, with maternal mortality ratio of around 270 per 100,000 live births and neonatal mortality rate of around 40 per 1000 live births [34, 35]. According to government data, 30–40% of the population comprise of fishermen and the community is a hub for gutka manufacturing and distribution [36].

## Sample size

A sample size of at least 510 women was estimated to detect a prevalence difference of 12% with an alpha of 0.05 and 80% power among gutka users and non-gutka users. An inflated sample size of 557 women was used after taking into account 8.4% of dropouts or refusals after informed consent. Based on the antenatal services provided by the study site, the dropout rate is calculated. OpenEpi software was used to calculate the sample size. For the qualitative phase, a purposive sampling technique was adopted from the same cohort who participated in the survey.

## Sampling technique and eligibility

Randomly selected MWRA were approached and screened for eligibility for participation using available line listings from the existing maternal and child health services in the community. Non-pregnant MWRA with aged 18–49 years who were permanent resident of the catchment areas, did not have a birth in 6 months prior and provided written consent to participate were enrolled. MWRA with any known diagnosed blood disorder for example Thalassemia, sickle cell anemia and any type of blood cancer were excluded.

## Ethics considerations

Ethical approval was obtained from the Ethical Review Committee at Aga Khan University (reference number: 5090-CHS-ERC-17). Written informed consent was administered in a local language to eligible participants after an explanation of the research. In cases where MWRA were not unable to read or write, consent was documented by a thumbprint in the presence of literate witnesses.

## Informed consent procedure and confidentiality

Among the eligible participants, written informed consent was obtained by the research team in a local language. Every participant was given a unique study ID and anonymity of the participant was maintained. Data confidentiality was assured as all data was stored in a locker and only accessible to the study team. All the anemic participants were given nutritional counselling later and, where required, were referred to the primary health clinic for a free-of-cost consultation with a physician.

## Data collection tools and variables

A screening tool was created to assess the eligibility criteria among those who were approached. A 'structured questionnaire' was developed gleaning information on demographics, household-related information, obstetric history, socio-economic status, information on gutka consumption, food frequency according to four-week recall and food insecurity at the household level.

**Household demography.** A variety of demographic variables were collected, such as the age of the women, ethnicity, number of people living in the household, education, and employment status. Information on households is collected using variables such as the number of people in the household, the condition of the house, the type of construction, the fuel used to cook, the source of drinking water, the type of toilet, and standard questions about household assets. These questions are then converted into proxy indicators for wealth quintiles, i.e., the richest, the middle, the poor, and the poorest of the poor. Previously conducted surveys were used to develop these variables [37, 38].

**Food diversity.** For food diversity, a list of standard iron-rich foods is created based on common availability in Pakistan, using a food frequency questionnaire [39]. The items included red meat (beef, mutton, and lamb), organs (liver, kidney), poultry meat, fresh and dry fruits, green leafy vegetables, legumes, red beans, and chickpeas. In addition to seafood consumption, fish and prawn consumption were also asked about. The frequency of consumption is measured for each variable using standard responses such as 'don't eat at all', 'once a month to twice a month', and 'at least once a week'. Following discussion with community representatives and community health workers, the variable for seafood was added. It was discovered that White Pomfret, Shad, Arabian Sea Meagre, and Barramundi and Croaker fish were the most consumed marine fishes. In the fishing community, shrimp is also cooked in the majority of households. The iron content of most of these seafood items is less than 1.5 mg per serving. A monotonous diet variable is created by deep diving into data sets (quantitative and qualitative), which has the operation definition of only consuming seafood without any other major sources of iron-containing foods such as red meat, and other major food items.

**Gutka consumption.** Specifically, for gutka, data was collected on past and current consumption as daily or less than daily if taken, adapted from tobacco survey questions [40]. Additionally, if participants reported to be consuming gutka, data was collected on the number of packs and type of gutka they consume.

**Food insecurity.** Food insecurity questions were adopted from the 'Household Food Insecurity Access Scale for Measurement of Food Access: Indicator Guide Version 3 (HFIAS)' [41]. A total of 18 questions were asked about four different themes of food insecurity at the household level, including household access to food insecurity (Conditions), household access to food insecurity (Domains), household access to food insecurity (Scale Score) and household access to food insecurity (Prevalence) [41].

**Community experiences.** Further, 'semi-structured guide' for IDIs and FDGs was designed to get the community insight and a checklist was created to assess the availability of iron rich food items in the community. IDIs and FDGs were conducted in a private, quiet room at the field office VITAL Pakistan Trust. These sessions were tape recorded, transcribed, translated, and transferred to the sheet for data analysis. Each FGD and IDIs lasted for 50–60 minutes. Further, for market survey 14 different clusters were identified at study setting having different ethnicities. In the clustered, shops/stores/carts of interest were identified and after permission, each of these were visited 4 times in a week for 4 weeks to capture data on pattern of availability of food items, such as meat, poultry, fruits, and vegetables which are iron rich resources. The tools were developed in English language and translated into local language.

## Data collection

The data collection commenced from April 10, 2018, to May 20, 2018. Interviews were conducted by the trained community health workers for interview and blood sample collection with one senior research assistant. The interviews were conducted in local language followed by a collection of blood sample. Each interview lasted for approximately 30–40 minutes. All the Interviews and blood sample collection were carried out at the homes of the women in privacy. Senior research staff maintained the quality of data by field and office editing. In case of missing data, the participants were revisited very next day by the team for further verification and completeness. The FGDs and IDIs were conducted using a interview guide, tape recorded, transcribed, and translated into English for data analysis. Observation visits were made at shops to understand the community day to day type of food availability and purchasing practices using a checklist.

## Specimen collection, transportation, and reporting

For hemoglobin assessment, the cyanmethemoglobin technique was used in the laboratory, and venous blood was drawn during specimen collection. The data collection team was accompanied by trained phlebotomists who collected specimens on a daily basis. In accordance with WHO Guidelines on Drawing Blood: Best Practices in Phlebotomy, they were trained on standard operating procedures for blood specimen collection [42]. The samples were transported to Koohi Goth Hospital Laboratory Research and Training Center, Karachi using tubes and carrier boxes at room temperature following the proper procedure of transportation. An assay for complete blood count (CBC) was run on the same day. Following approval from the pathology head, final reports were delivered to the research team the next day. Team members provided a copy of the report to the family, which contained complete CBC indicators.

## Data analysis

Data analysis was conducted using Stata version 16. Descriptive analyses were carried out for basic demographic covariates and are presented as frequencies and percentages. Using the international analysis guideline of the HFIAS, key indicators such as: households experiencing condition at any time during the recall period, households experiencing condition at a given frequency, household food insecurity access-related domain, HFIAS scores, average HFIAS scores, HFIA categories, HFIA prevalence, and food security level of each household were calculated [41]. Binary categories of food frequency and food insecurity variables were created for the final model. In light of the fact that the questionnaire captures information about different types of food items, the qualitative findings helped us develop variables of monotonous diets, which are defined in this study as seafood (fish and/or shrimp) as the sole source of meat, and red meat includes beef, mutton, chicken, liver, and kidney. Principal Component Analysis (PCA) was used to calculate wealth quintiles [38]. Data on productive and non-productive assets as well as household utilities and other were given a factor weight. A factor weight was applied to the data on productive and non-productive assets, household utilities, and other factors. In order to calculate wealth indexes, factor weights were applied to the components. A wealth index score for each of the five quintiles was generated after that.

For the main statistical analysis, unadjusted/crude odds ratios (CORs) with 95% confidence intervals (CIs) were calculated in the univariate analysis. Variables with a p-value less than 0.25 in the univariate analysis were entered into the stepwise multivariate logistic regression analysis, and adjusted odds ratios (AORs) were calculated. AORs with a significance level of less than 0.05 are reported. Complete-case analysis was done, and interaction was checked between all plausible variables in which no interaction was found. Further, no multi-collinearity was found between any variables. Based on dietary patterns reported, a variable of monotonous dietary consumption was created which is related to the consumption of only seafood as the main source of daily food source. For qualitative data, manual thematic analysis was performed. Data obtained from the audio tapes was transcribed into scripts verbatim in word document. Using an inductive thematic approach, data (quotes) were examined for recurrent instances. Then systematically identified across the data set and grouped together by means of a coding system [43].

# Results

## Cross-sectional survey

A total of 600 non-pregnant MWRA (18–49 years) were screened for eligibility to reach the sample size during April 10 and May 10, 2018. After screening process, 98.2% (n = 589) met

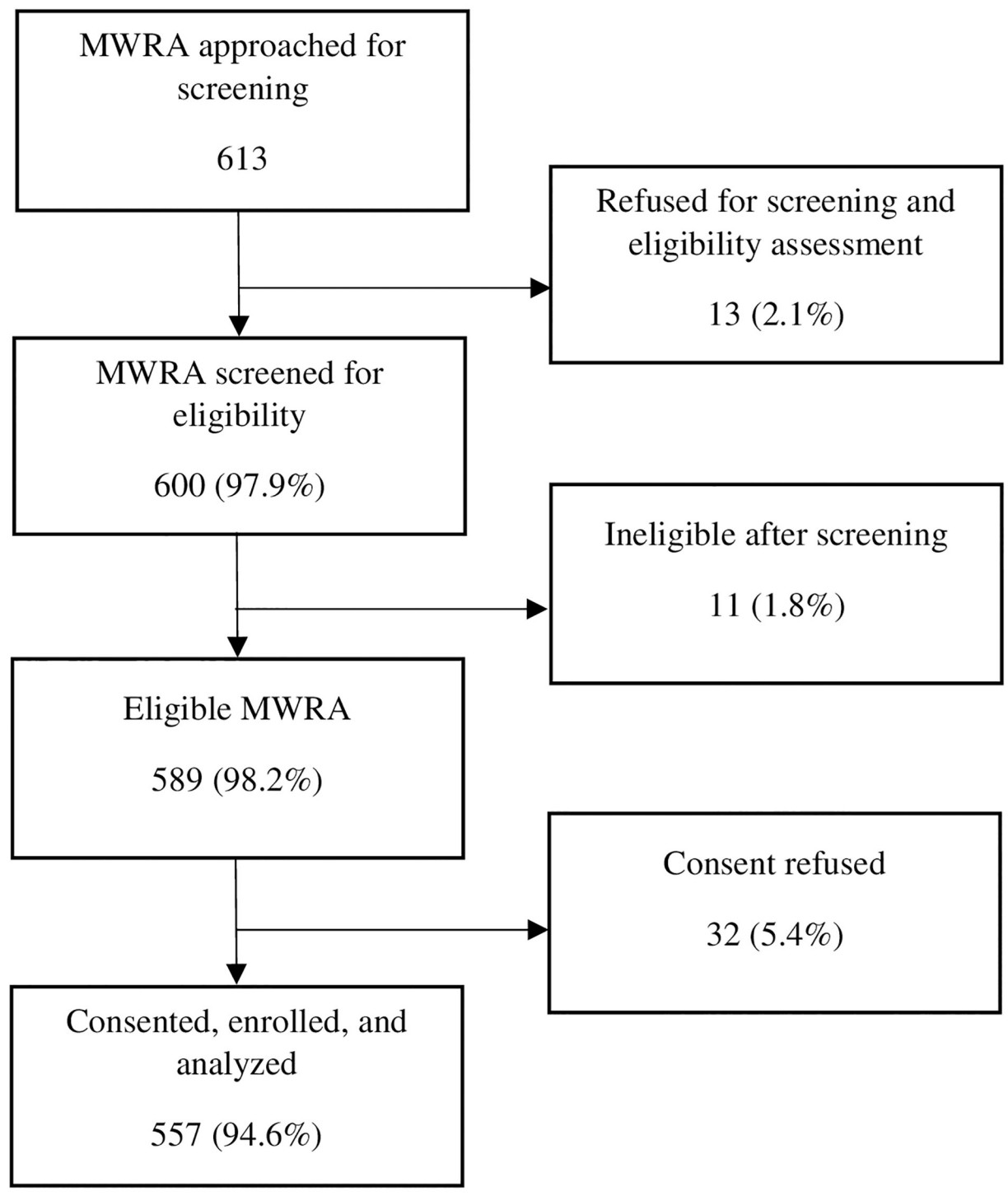

**Fig 1. Screening, eligibility, and consent.**

the inclusion criteria, and 2.8% (n = 17) refused to provide give consent for specimen, 2.6% (n = 15) refused to give consent for overall study, and 94.6% (n = 557) were consented and enrolled (Fig 1). The overall prevalence of anemia among MWRA was 68.0%. The prevalence of mild anemia (hemoglobin: 11.0 to 11.9 gm/dl) was 33.2%, moderate anemia (hemoglobin:

**Table 1. Baseline characteristics.**

| Characteristics | Values |
|---|---|
| **Anemia status—n (%)** | |
| Overall anemia | 379 (68.0) |
| *Mild anemia* | 185 (33.2) |
| *Moderate anemia* | 169 (30.3) |
| *Severe anemia* | 25 (4.5) |
| No anemia | 178 (32.0) |
| **Complete blood count panel—Mean ± SD** | |
| Hemoglobin (gm/dl) | 11.27 ± 1.65 |
| Red Blood Cells | 4.50 ± 0.64 |
| Hematocrit | 34.50 ± 4.49 |
| Mean Corpuscular Volume | 78.66 ± 35.78 |
| Mean Corpuscular Hemoglobin | 25.14 ± 4.31 |
| Mean Corpuscular Hemoglobin | 32.18 ± 1.93 |
| **Maternal characteristics** | |
| Age in years—Mean ± SD | 31.3 ± 7.9 |
| Age in years—Median (IQR) | 30 (25–37) |
| **Ethnicity—n (%)** | |
| Sindhi | 417 (74.9) |
| Pashtun | 56 (10.1) |
| Urdu | 42 (7.5) |
| Balochi | 37 (6.6) |
| Others | 5 (0.9) |
| **Education–n (%)** | |
| No formal education | 378 (67.9) |
| Primary | 98 (17.6) |
| Secondary and above | 81 (14.5) |
| **Family structure—n (%)** | |
| Nuclear | 337 (60.5) |
| Joint | 220 (39.5) |
| **Obstetric history–Median (IQR)** | |
| Gravidity | 4 (2–6) |
| Parity | 3 (1–5) |
| **Gutka patterns–Median (IQR)** | |
| Daily consumption | 2 (3) |

10.9 to 8.0 gm/dl) was 30.3% and severe anemia (hemoglobin: less than 8.0 gm/dl) was 4.5% of women. The mean hemoglobin level among the MWRA was 11.27 ±1.65 gm/dl. Further, the mean age of the MWRA was 31.3 ±7.9 years. Sindhi is the most frequently spoken language, i.e., 74.9% and 67.9% MWRA had no formal education (Table 1).

Overall, 59.43% of the households were mild to severe food insecure. Seafood intake was high among the participants, 77.20% of the women consumed seafood in the form of fish or prawn at least once in a month. Only 31.42% MWRA mentioned that consumption of red meat at least once in a month. Further, 77.20%, 79.54%, and 52.24% of MWRA consumed dairy products, poultry meat and eggs respectively. Iron rich vegetable are consumed by 88.33% of the MWRA. Among the participants 39.1% of MWRA reported to be consumed gutka (Table 2).

**Table 2. Food insecurity and pattern of iron rich food consumption pattern at household level (N = 557).**

| Variables | Value |
|---|---|
| **Household Food Insecurity Access Scale (HFIAS) for Measurement of Food Access—n (%)** | |
| Percent of households that ran out of food—condition | 237 (42.55) |
| Percent of households that ran out of food often—condition | 98 (17.59) |
| Percent of households with insufficient food quality—domain | 320 (57.45) |
| **Food Insecurity Access Scale Score–n** | |
| HFIAS Score | 4707 |
| Average HFIAS Score | 8.45 |
| **Food insecurity prevalence—n (%)** | |
| Food insecure (total) | 331 (59.43) |
| *Mild* | 45 (8.08) |
| *Moderate* | 42 (7.54) |
| *Severe* | 224 (43.81) |
| Food secure | 226 (40.57) |
| **Food diversity—n (%)** | |
| *Seafood consumption* | |
| At least one time per week per day | 148 (26.57) |
| 1 to 2 times per month | 282 (50.63) |
| Do not eat at all | 127 (22.80) |
| *Red meat consumption* | |
| At least one time per day per week | 64 (11.49) |
| 1 to 2 times per month | 111 (19.93) |
| Do not eat at all | 382 (68.58) |
| *Dairy product consumption* | |
| At least one time per day per week | 81 (14.54) |
| 1 to 2 times per month | 349 (62.66) |
| Do not eat at all | 127 (22.80) |
| *Poultry consumption* | |
| At least one time per day per week | 182 (32.68) |
| 1 to 2 times per month | 261 (46.86) |
| Do not eat at all | 114 (20.47) |
| *Egg consumption* | |
| At least one time per day per week | 130 (23.34) |
| 1 to 2 times per month | 161 (28.90) |
| Do not eat at all | 266 (47.76) |
| *Legume consumption* | |
| At least one time per week per day | 289 (51.89) |
| Do not eat at all | 268 (48.11) |
| *Iron rich vegetables* | |
| At least one time per week per day | 185 (33.21) |
| 1 to 2 times per month | 307 (55.12) |
| Do not eat at all | 65 (11.67) |
| **Gutka consumption—n (%)** | |
| No | 339 (60.86) |
| Yes | 218 (39.14) |

Further, among the covariate maternal education, gutka consumption, red meat intake and monotonous diet were found to be significant predictors of anemia on multivariate regression analysis. Approximately 75%. Of MWRA with no formal education were anemic (AOR: 2.93 95% CI: 1.90–4.52). Of those women who were consuming Gutka, 81.7% were anemic (AOR: 2.84 95% CI: 1.81–4.46). More that 68% of women were not eating any kind of meat and of these 74.0% were anemic (AOR: 1.68 95% CI: 1.06–2.65). There were 16.2% of MWRA who reported monotonous diet consumption, all of them were from fisherman community and 90% of them were anemic (AOR: 4.56 95% CI: 1.38–15.02) (Table 3).

## Qualitative insights

The thematic analysis of qualitative component revealed enriching information at household and community level which further support quantitative findings in reaching to the root cause of anemia in this community. The framework based on community insights is presented in Fig 2.

**Household environment.** *Individual.* Most of the women in the community were not aware of food items rich in the iron content. Also, they were not aware of the harmful effects of using gutka on their health. In one of the focus groups discussion one woman mentioned.

"We don't know what the sources of iron (folaad) in our food are. I feel that it might be meat."

(35 years old MWRA from Qasmani para)

Women belonging to the fishermen community believed that seafood is a good source of food for health and provides all sort of nutrition required.

"Fish and prawn are the main food items we use. It contains all we need. It is good enough."

(26 years old MWRA from AminJatt para)

*Family.* This fishermen community has a patriarchal structure where preference is given to the male of the household and main decisions are also taken by men. However, for a day-to-day food menu, a woman and her husband, and other family members had an equal opportunity to decide about daily menu, however, with limited options based on socioeconomic status the choice of food becomes very limited. One woman commented:

"We have limited food, what to decide among the limited options, like fish and prawn. . . this is what we all decide to cook. . . . . ."

(31 years old MWRA from Milkaye para)

In fisherman community, most of the time husbands are out at the sea and woman is the main decision maker in selecting the food items with limited available resources, as mentioned by one of the participants as:

"My husband and other male members at home are fishermen, they are at sea most of the time, and so we cook whatever is available to us. . ..

(27 years old MWRA from Moosani para)

**Table 3. Predictors of anemia among non-pregnant MWRA—Crude and adjusted odd ratio (N = 557).**

| Variables | Total n (%) ⁋ | Prevalence of anemia n (%) ‡ | Crude odds ratio (95% CI) | P-value | Adjusted odds ratio (95% CI) | P-value |
|---|---|---|---|---|---|---|
| **Maternal characteristics** | | | | | | |
| **Age** | | | | | | |
| Less than 25 | 159 (28.55) | 102 (64.15) | Ref. | | | |
| Greater than 25 and less than 35 | 249 (44.70) | 168 (67.47) | 1.16 (0.76–1.76) | 0.49 | | |
| Greater than 35 | 149 (26.75) | 109 (73.15) | 1.52 (0.93–2.47) | 0.09 | | |
| **Ethnicity** | | | | | | |
| Other Languages | 140 (25.13) | 291 (69.78) | Ref. | | | |
| Sindhi Language | 417 (74.87) | 88 (62.86) | 1.36 (0.91–2.03) | 0.129 | | |
| **Gravidity** | | | | | | |
| Less than equals to 2 | 207 (37.16) | 128 (61.83) | Ref. | | | |
| Greater than 2 | 350 (62.84) | 251 (71.71) | 1.56 (1.08–2.25) | 0.016 | | |
| **Parity** | | | | | | |
| Less than equals to 2 | 246 (44.17) | 154 (62.60) | Ref. | | | |
| Greater than 2 | 311 (55.83) | 225 (72.34) | 1.56 (1.09–2.23) | 0.015 | | |
| **No of children alive** | | | | | | |
| Less than equals to 2 | 264 (47.40) | 165 (62.5) | Ref. | | | |
| Greater than 2 | 293 (52.60) | 214 (73.03) | 1.62 (1.13–2.32) | 0.008 | | |
| **Number of children under-five years** | | | | | | |
| Less than equals to 2 | 219 (39.32) | 148 (67.57) | Ref. | | | |
| Greater than 2 | 338 (60.68) | 231 (68.34) | 1.03 (0.71–1.49) | 0.85 | | |
| **Number of children under-two years** | | | | | | |
| Less than equals to 2 | 348 (62.48) | 233 (66.95) | REF | | | |
| Greater than 2 | 209 (37.53) | 146 (69.85) | 1.14 (0.78–1.65) | 0.477 | | |
| **Women Education** | | | | | | |
| At least Primary | 179 (32.14) | 95 (53.07) | Ref. | | | |
| No formal education | 378 (67.86) | 284 (75.13) | 2.67 (1.83–3.88) | <0.00 | 2.93 (1.90–4.52) | <0.00 |
| **Woman's Occupation** | | | | | | |
| Employed | 40 (7.18) | 25 (62.5) | Ref. | | | |
| Unemployed | 517 (92.82) | 354 (68.4) | 1.30 (0.66–2.53) | 0.436 | | |
| **Gutka consumption** | | | | | | |
| No | 339 (60.86) | 201 (59.29) | Ref. | | | |
| Yes | 218 (39.14) | 178 (81.65) | 3.05 (2.03–4.58) | <0.00 | 2.84 (1.81–4.46) | <0.00 |
| **Gutka consumption in past** | | | | | | |
| No | 23 (4.13) | 17 (73.91) | Ref. | | | |
| Yes | 534 (95.87) | 362 (67.79) | 1.34 (0.52–3.47) | 0.539 | | |
| **History of worm infestation** | | | | | | |
| No | 43 (7.72) | 31 (72.09) | Ref. | | | |
| Yes | 514 (92.28) | 348 (67.70) | 1.23 (0.61–2.46) | 0.554 | | |
| **Modern contraceptive use** | | | | | | |
| Yes | 202 (36.27) | 133 (65.84) | Ref. | | | |
| No | 355 (63.73) | 246 (69.29) | 1.17 (0.81–1.69) | 0.401 | | |
| **Household Characteristics** | | | | | | |
| **Family structure** | | | | | | |
| Joint system | 220 (39.50) | 145 (65.90) | Ref. | | | |
| Nuclear system | 337 (60.50) | 234 (69.43) | 1.17 (0.81–1.68) | 0.383 | | |
| **Husband occupation** | | | | | | |
| Employed non-fisherman | 305 (54.76) | 209 (68.52) | Ref. | | | |

*(Continued)*

**Table 3.** (Continued)

| Variables | Total n (%) ¶ | Prevalence of anemia n (%) ‡ | Crude odds ratio (95% CI) | P-value | Adjusted odds ratio (95% CI) | P-value |
|---|---|---|---|---|---|---|
| Fishermen | 212 (38.06) | 145 (68.39) | 0.99 (0.68–1.44) | 0.975 | | |
| Unemployed | 40 (7.18) | 25 (62.5) | 0.76 (0.38–1.51) | 0.444 | | |
| **Toilet facility at the household** | | | | | | |
| Flush toilet | 443 (79.53) | 287 (64.78) | Ref. | | | |
| Pit toilet | 114 (20.47) | 92 (80.7) | 2.27 (1.37–3.76) | 0.001 | | |
| **Regular fecal disposal facility available** | | | | | | |
| Yes | 485 (57.27) | 319 (65.77) | Ref. | | | |
| No | 72 (10.77) | 60 (83.33) | 2.60 (1.36–4.97) | 0.004 | | |
| **Wear shoes before using toilet** | | | | | | |
| Yes | 507 (60.32) | 336 (66.27) | Ref. | | | |
| No | 50 (7.72) | 43 (86) | 3.12 (1.37–7.09) | 0.006 | | |
| **Handwashing with soap after toilet use** | | | | | | |
| Yes | 437 (53.50) | 298 (68.19) | Ref. | | | |
| No | 120 (14.54) | 81 (67.5) | 0.96 (0.62–1.49) | 0.885 | | |
| **Wealth quantile** | | | | | | |
| Richest | 74 (13.29) | 37 (66.66) | Ref. | | | |
| Rich | 76 (13.64) | 35 (68.46) | 1.08 (0.61–1.90) | 0.774 | | |
| Middle | 71 (12.75) | 41 (63.39) | 0.86 (0.49–1.50) | 0.608 | | |
| Poor | 78 (14.00) | 33 (70.27) | 1.18 (0.67–2.08) | 0.564 | | |
| Poorest | 80 (14.36) | 32 (71.42) | 1.25 (0.70–2.20) | 0.442 | | |
| **Household Food security*** | | | | | | |
| Food secure | 226 (40.57) | 73 (67.69) | Ref. | | | |
| Food insecure | 331 (59.43) | 105 (68.27) | 1.02 (0.71–1.47) | 0.886 | | |
| **Household food pattern for iron-rich diet** | | | | | | |
| **Sea food consumption†** | | | | | | |
| Does not consume seafood | 127 (22.80) | 83 (65.35) | Ref. | | | |
| Consumes seafood | 430 (77.20) | 296 (68.83) | 1.17 (0.77–1.77) | 0.46 | | |
| **Red meat consumption** | | | | | | |
| Consumes red meat | 175 (31.42) | 96 (54.85) | Ref. | | | |
| Does not consume red meat | 382 (68.58) | 283 (74.08) | 2.05 (1.41–2.97) | <0.00 | 1.68 (1.06–2.65) | 0.026 |
| **Dairy products consumption** | | | | | | |
| Consumes dairy | 430 (77.20) | 297 (69.06) | Ref. | | | |
| Does not consume dairy | 127 (22.80) | 82 (64.56) | 0.81 (0.53–1.23) | 0.339 | | |
| **Poultry consumption** | | | | | | |
| Consumes poultry | 443 (79.53) | 283 (63.88) | Ref. | | | |
| Does not consume poultry | 114 (20.47) | 96 (84.21) | 3.01 (1.75–5.17) | <0.00 | | |
| **Eggs consumption** | | | | | | |
| Consumes eggs | 291 (52.24) | 200 (68.72) | Ref. | | | |
| Does not consume eggs | 266 (47.76) | 179 (67.29) | 0.93 (0.65–1.33) | 0.717 | | |
| **Dry fruits** | | | | | | |
| Consumes fruits | 89 (15.98) | 61 (68.53) | Ref. | | | |
| Does not consume fruits | 468 (84.02) | 318 (67.94) | 0.97 (0.59–1.58) | 0.913 | | |
| **Legume's consumption** | | | | | | |
| Consumes legumes | 289 (51.89) | 194 (67.12) | Ref. | | | |
| Does not consume legumes | 268 (48.11) | 185 (69.02) | 1.09 (0.76–1.55) | 0.631 | | |

(*Continued*)

**Table 3.** (Continued)

| Variables | Total n (%) ⁕ | Prevalence of anemia n (%) ‡ | Crude odds ratio (95% CI) | P-value | Adjusted odds ratio (95% CI) | P-value |
|---|---|---|---|---|---|---|
| **Vegetable's consumption** | | | | | | |
| Consumes vegetables | 492 (88.33) | 331 (67.27) | Ref. | | | |
| Does not consume vegetables | 65 (11.67) | 48 (73.84) | 1.37 (0.76–2.46) | 0.287 | | |
| **Monotonous diet consumption**** | | | | | | |
| No | 467 (83.84) | 298 (63.81) | Ref. | | | |
| Yes | 90 (16.16) | 81 (90) | 5.10 (2.49–10.42) | <0.00 | 4.56 (1.38–15.02) | 0.012 |

⁕ Percentage calculated out of total sample size, i.e., 557

‡ Prevalence of anemia is calculated using the total of specific categories as a denominator.

* Food insecurity is derived by merging mild, moderate and severe HFIA categories.

† Reference category of "not consuming seafood" is selected based on the assumption that mostly fishermen's household consumes seafood as the main source of meat, which may lead to anemia.

** Monotonous diet defined as only consuming seafood with no other major source of iron rich food items. These 90 participants are from the fishermen community.

Gender preference in serving the meal first to the male members is common in the community and perceived as a common custom.

"I always serve my husband first, the best of curry and fresh Roti, this is my duty and I follow that. I eat in the end. . ."

(28 years old MWRA from Lalabad)

Another woman shared:

**Fig 2. Updated framework of high gutka consumption and low iron rich diet in coastal slums.**

"I eat in the end, after serving my husband and family members, this is our family tradition, and we have to serve them first."

(31 years old MWRA from Moosani para)

*Gutka consumption among females.* Gutka is a form of tobacco with some element of recreational substance in it. Its use is very common in men, women, and children especially among poor households. It gives a sense of well-being and suppresses hunger. Many women mentioned that gutka has addiction, which make them eat it on a regular basis. One of the women from FGD who was eating gutka, said:

"I like to eat it all the time, I like it! The gutka make me feel better, it just 10–20 Rs and cost of food is high, I like to buy it. . ."

(31 years old MWRA from Qasmani para)

Another woman highlighted:

"After eating gutka, I don't feel like I am hungry, maybe this is the reason I am very weak. We are so poor, at least this helps us. . ."

(31 years old female from Dabla para)

"We shared gutka with each other. Around my home, everyone chews gutka. I don't remember how I started it." (46 years old MWRA from Milkaye para)

In one of the FGDs with women from the fishermen community a woman mentioned that gutka consumption is increasing day-by-day and people are giving it priority over food. One woman commented

"This thing has become an addiction now a day, we can survive without meat and fruits and other things you mentioned, what you call? Yeah, iron rich diet; we can't survive without gutka."

(39 years old MWRA from Dabla para)

"I preferred to eat Rehri gutka, it suits me, and my husband also likes to eat the same, this is very readily available and made in community"

(40 years old female from Qasmani Para)

Another woman shared:

"I am using it (gutka) for a very long time, and I know that I can easily get it (gutka), but cannot afford meat"

(40 years old MWRA from Chasma goth)

**Low household income.** The other common factor at household level with limited food choices is poverty, which may impact in making the choice for other expensive food items.

"We are poor fishermen. We cannot afford meat and things such as that to eat. We can only eat what we get free, the fish and prawn. This is only available free food for my home. Gutka, I can easily buy with very less amount of money"

(31 years old MWRA from AminJatt para)

**Community environment.** *Fishermen.* In this community, meat is seldom consumed. Meat is not sold regularly in the markets and is very expensive and most of the families could not purchase it on a regular basis. People generally prefer fish and prawns because these items are easily available in the fishing community, they are cheap and mostly are their own catch.

"In my village, I don't see anyone selling meat, which you are talking about. It's all fish we like. We don't like the taste of meat; it has some smell, which we don't like. I guess therefore no meat shop is here."

(36 years old MWRA from Moosani para)

*Availability of iron rich diet.* There were very few meat shops in the community bazars. According to some accounts, the issue is not that of supply but rather that of demand.

"In our community, there is no meat shop. People don't like to eat meat; no one wants to open the shop where people are not willing to buy meat. At Eid (Eid-ul-Adha, a religious festival), people sacrifice cattle; this is the only time when we eat meat. Even, many don't like the taste though."

(24 years old midwife)

"There is no meat shop in this whole village. No one wants to sell it because the market is not good here. See, I have only a few chickens available; very rarely someone comes to buy it also. Meat has no market."

(Poultry shopkeeper from Moosani para)

"See, meat is 800–900 Rupees per Kg and mutton is even more expensive, what do you think, can these people afford it? No! They cannot. Look at the cost of Mangoes, 120 to 200 Rupees per kg. No can buy these expensive items if they have many members at home."

(A fruit cart owner at Jabal)

## Findings from market survey

A total of only 3 meat shops were identified at the study site and beef as a red-meat source was available on 3 out of 12 observations on these shops. Total 9 poultry shops were found, and availability of poultry meat was observed on 23 times out of 36 observations made (Table 4).

When experiences of shopkeepers were explored, it was identified that red meat has very limited availability in the community. The meat shops are hardly any in the fisherman community and if someone must buy a meat, the only option is going to the main city. Many of the shopkeepers related this to the cost and highlighted that community has limited purchasing power to buy expensive food which ultimately affect the consumption of food variety. One shop keeper commented

**Table 4. Findings of market survey.**

| Food items | Geographical visited for observation | Availability of specific shop in the area | Observations at each available shop [a] | Availability of specific iron-rich food item |
|---|---|---|---|---|
| | N | N (%) | N | N (%) |
| **Meat shops** | | | | |
| Beef | 14 | 3 (21.4) | 12 | 5 (41.7) |
| Mutton | | | 12 | 0 (0.0) |
| Lamb | | | 12 | 0 (0.0) |
| Liver | | | 12 | 1 (8.3) |
| **Poultry shops** | | | | |
| Chicken meat | 14 | 9 (64.3) | 36 | 23 (63.9) |
| Liver | | | 36 | 23 (63.9) |
| **Fruits shops/carts** | | | | |
| Dates | 14 | 8 (57.1) | 32 | 13 (40.6) |
| Apricot | | | 32 | 12 (37.5) |
| Prunes | | | 32 | 0 (0.0) |
| Apple | | | 32 | 21 (65.6) |
| Watermelon | | | 32 | 27 (84.4) |
| Strawberries | | | 32 | 8 (25.0) |
| Figs | | | 32 | 11 (34.4) |
| Pomegranate | | | 32 | 7 (21.9) |
| **Major green vegetable shops/cart** | | | | |
| Spinach | 14 | 11 (78.6) | 44 | 28 (63.6) |
| Mustard leaf | | | 44 | 8 (18.2) |
| Bitter gourd | | | 44 | 8 (18.2) |
| Indian squash | | | 44 | 3 (6.8) |
| Capsicum | | | 44 | 3 (6.8) |
| Qasoori methi | | | 44 | 1 (2.3) |
| Moongray | | | 44 | 4 (9.1) |
| Green pea | | | 44 | 0 (0.0) |
| Beetroot | | | 44 | 3 (6.8) |
| **Grocery shop** | | | | |
| Red beans | 14 | 13 (92.9) | 52 | 2 (3.8) |
| Chickpeas | | | 52 | 43 (82.7) |
| Lentil | | | 52 | 47 (90.4) |
| Eggs | | | 52 | 48 (92.3) |
| **Pan shop** | | | | |
| Gutka | 14 | 14 (100.0) | 56 | 56 (100.0) |

[a] Each shop was visited four times on a weekly basis during the one-month period to capture in-depth data on patterns,

"There is no meat shop in this area, no one wants to sell it because the market is not good here, see I have only few chickens available, very rarely someone comes to buy it also, meat has no market." (Poultry shop in Moosani para)

## Discussion

The root causes of anemia in women living in Rehri Goth, a coastal slum of Karachi were poverty, habitual intake of gutka, food insecurity and lack of diversity of food. The quantitative

data is well supported by the qualitative data which revealed in-depth information on the sufferings of this poor population. The rates of anemia identified in our study are higher compared to overall prevalence in Sindh province as well as its largest city Karachi [30]. These finding are consistent with some regional data showing higher rates of anemia in the populations living in the coastal slums mainly for households which are food insecure, and source of income is fishing [44–47].

In our multivariate model wealth index was not significantly associated with anemia. This was mainly due to an artificial division of an already poor population into wealth quintiles which are not truly representative of the ground realities. An alternate coding of the wealth quintiles in such areas could be poorest of the poor, poorer, poor, and least poor. The facts distinctly surfaced during focus group discussions and in-depth interviews of shop keepers where poverty and lack of purchasing power of the community were identified as main factors for not having access to diverse food and low demand for other sources of meat and fruits as close to 60% of the households were identified as food insecure inn this area. The seasonality of earning among fishermen households may link poor access to quality diet [10, 19, 47]. Risk of anemia was significantly higher among women who never consumed red meat and among those who used only seafood in diet. Even high consumption if seafood as monotonous diet among those from the participants, still resulted in anemia which is potentially because of iron in the type of fish or shrimp consumed [48]. The poor accessibility to red meat in this study is much higher than the population enrolled in a national nutrition survey in Pakistan which suggested that only 13.7% of the population does not eat red meat at all. The market survey also highlighted the key issue of low demand of meat and poultry compared to seafood especially in the in the clusters where fishermen live. This is a common phenomenon because seafood brought to the market is their own catch [19] and mostly household members eat leftovers from the fishing market. Theoretically, fishermen have access to seafood, which is their own catch, but households only consume leftovers, a small share they receive after selling it to consumers [19]. Moreover, fishing in this and other neighboring slums is highly dependent on the seasons. The money received by fishermen after they sell their catch is also dominated by coastal feudal lords [49]. Poverty in such areas is linked with poor literacy rate hindering opportunities to health awareness, options for better life and for that matter poor realization of the importance of healthy diet [10, 19, 33]. In communities like this, households may opt to have small-scale poultry farming, but this is not very common at the study site. Poultry is an excellent source of protein, and eggs can provide macro- and micronutrients. In contrast, only 15% of respondents indicated that they have poultry at their homes, mostly in non-fisherman's communities. Cattle farming is also rare in the community. It is also difficult and expensive to find other sources of meat in a community like this [19]. Therefore, high consumption is caused by both community preferences and high meat prices in the country.

The use of gutka is an emerging health risk in the developing world [50]. South Asians have the highest prevalence of gutka consumption, with an increasing trend among women; the health risks associated with its use are often ignored [51]. The use of gutka among women residing in slums is strongly associated with poverty, knowledge, and education status [52]. This study suggests that the use of gutka at the study site is a strong predictor of anemia, which may have significant importance at other coastal slums too. A premise for including gutka consumption is that its ingredients may hinder intestinal absorption and alter other biological pathways. One of the ingredients in chewable tobacco is iron, which has effects on iron metabolism, hemoglobin levels, and reservoirs [24–26]. There is a possibility that this alters or suppresses hepcidin mRNA expression and reduces hepcidin concentration in pregnant women, which is necessary for iron homeostasis [24–26]. In addition, crushed areca nut contains alkaloids that also affect intestine's ability to absorb iron [27–29]. Calcium hydroxide or slake lime

also inhibits iron absorption. It is possible that the correlation between gutka and anemia in this study favors the same pathways. A more robust study is needed to establish this fact, which was beyond the scope of this study. Furthermore, gutka suppresses hunger and gives a false sense of wellbeing [52], Although there is no nutrition value reported in the literature. Study participants reported similar experiences. This study is not able to show significance of other predictors of anemia like gravidity, parity, food insecurity, and contraception which are reported in literature.

## Strength and limitations

Despite its small scale, this study highlighted key issues pertaining to maternal health at Rehri Goth. By triangulating the data using mix-method approaches, we have added the flavor of experience and learning from the field, thereby strengthening our findings. By generating evidence, other researchers may be able to study the topic with more robust study designs in the future to delve deeply into the issue. The measurement of hemoglobin is chosen as the indicator for anemia. This is invasive, and its collection is often problematic from a practical and ethical standpoint at the household level. The study has successfully collected all specimens under supervision and monitoring [53]. However, there are other biomarkers that are specific to the diagnosis of iron deficiency anemia, such as Transferrin and Ferritin. But they require more advanced laboratory testing and are more expensive. Hence, measuring blood hemoglobin is a more reliable indicator of anemia at the population level, and it is also more cost-effective. It was difficult to collect quality data on the proportion and size of meals and snacks during the study, which was one of the limitations of the study. We faced two challenges in measuring the size of meals or snacks. First from a cultural standpoint people feel sensitive about stating how much they consume. Second from a practical standpoint many found it difficult to assess. Similar challenges are reported in other studies [54]. In addition, the recall period of each food item is based on self-reports, but training of the staff with pictorial presentations of each food item and the use of local language has made it feasible [54]. The study was not able to identify the ingredients of all the gukta consumed by participants (around 11 types). Therefore, in this paper, only the main ingredients reported in the literature are discussed. As a final note, the average temperature during April and May is between 32 and 35 degrees Celsius. Based on past research experience in this area, data collection and results were not impacted. One could, however, conduct a longitudinal observational study to gain a deeper understanding of the impact of seasons.

## Conclusion and recommendations

Availability and accessibility of iron-rich food items is scared at Rehri goth and MWRA in this community have adopted the behavior of using a monotonous diet. Further, gutka consumption among MWRA has serious health consequences including anemia. Health promotion activities to provide counselling and awareness to MWRA about diversity in diet and discourage the use of gutka. Further, strategies may require in such communities to provide a safety net or low-cost utility stores by government to improve accessibility of iron-rich food, such as meat, and a strong commitment is needed to ban gutka use. There is currently no subsidy for meat products at these stores. However, there may be a potential opportunity to pilot this in impoverished communities. This requires political will and commitment from the provincial health department. A small-scale poultry or cattle farm can also be created to develop entrepreneurship and build local capacity. Women could be empowered and supported by initiatives like the Benazir Income Support Program (BISP) to create small businesses which could benefit their households and communities in accessing meat, especially in low-income areas.

When women are given the opportunity to participate in such initiatives, their health status improves [55, 56]. It is imperative to increase the purchasing power and affordability of healthy diets for the most vulnerable communities by implementing nutrition-sensitive social protection policies and initiatives [57]. It is also possible to improve behaviors using programs such as Lady Health Worker (LHW) as well as other horizontal and vertical health programs. By building frontline health worker capacity in this thematic area, which is currently very suboptimal in terms of gutka consumption, we can create awareness and change behaviors.

## Supporting information

**S1 Checklist. STROBE statement—Checklist of items that should be included in reports of cross-sectional studies.**
(DOC)

**S1 Dataset.**
(ZIP)

## Acknowledgments

The authors would like to thank the participants for their involvement, the research team for their contributions and the Faculty of Master of Science Epidemiology and Biostatistics program at Aga Khan University for guidance.

## Author Contributions

**Conceptualization:** Ameer Muhammad, Sarah Saleem.

**Formal analysis:** Ameer Muhammad.

**Investigation:** Ameer Muhammad.

**Methodology:** Ameer Muhammad.

**Resources:** Yasir Shafiq.

**Software:** Ameer Muhammad.

**Supervision:** Sarah Saleem.

**Validation:** Ameer Muhammad, Sarah Saleem, Yasir Shafiq.

**Writing – original draft:** Ameer Muhammad, Daniyaal Ahmad, Eleze Tariq.

**Writing – review & editing:** Ameer Muhammad, Sarah Saleem, Daniyaal Ahmad, Eleze Tariq, Yasir Shafiq.

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
