## [Decision Letter · Decision Letter 0]

16 Aug 2022

PONE-D-22-06122Gutka Consumption and Dietary Partialities Explaining Anemia in Women of a Coastal Slum of Karachi, Pakistan: A Mixed-method Study PLOS ONE

Dear Dr. Muhammad,

Thank you for submitting your manuscript to PLOS ONE. After careful consideration, we feel that it has merit but does not fully meet PLOS ONE’s publication criteria as it currently stands. Therefore, we invite you to submit a revised version of the manuscript that addresses the points raised during the review process.

Major Revisions 

We look forward to receiving your revised manuscript.

Kind regards,

Faisal Abbas, PhD

Academic Editor

PLOS ONE

Journal Requirements:

The author(s) received no specific funding for this work. However, the author received support from the VITAL Pakistan Trust through existing maternal, neonatal and child health programs to conduct laboratory assessments. 

7. Please include your tables as part of your main manuscript and remove the individual files. Please note that supplementary tables (should remain/ be uploaded) as separate "supporting information" files

Additional Editor Comments:

Major Revisions

Reviewers' comments:

Reviewer's Responses to Questions

**Comments to the Author**

1. Is the manuscript technically sound, and do the data support the conclusions?

Reviewer #1: Partly

Reviewer #2: Partly

2. Has the statistical analysis been performed appropriately and rigorously? 

Reviewer #1: Yes

Reviewer #2: I Don't Know

3. Have the authors made all data underlying the findings in their manuscript fully available?

Reviewer #1: Yes

Reviewer #2: No

4. Is the manuscript presented in an intelligible fashion and written in standard English?

Reviewer #1: Yes

Reviewer #2: Yes

5. Review Comments to the Author

Reviewer #1: PONE-D-22-06122

Gutka Consumption and Dietary Partialities Explaining Anemia in Women of a Coastal Slum of Karachi, Pakistan: A Mixed-method Study

In this paper the authors analyze the determinants of anemia in women in a coastal community of Pakistan. They analyze the role that gutka consumption and diet preferences play in women’s food intake decisions. The study utilizes a mixed- method framework and brings together important findings from the qualitative work to support the findings from the quantitative analysis. The results tables themselves were not available for me to review online, therefore my comments are based on the text in the manuscript.

Overall, I feel this is an under- studied question for a relatively less- researched geography. The mixed- methods approach is a strength, as is the fact that the study has information on biomarkers related to anemic. Having said that I feel there are important aspects wherein the paper can be strengthened, and I elaborate upon them below.

Placing the objectives against existing literature

It will be useful to sharpen the introduction in a way that focuses on a) what we know about anemia in Pakistan, b) what we know about nutritional outcomes of coastal communities and c) what we know about the relationship between Gutka consumption and health outcomes. The last point in particular is lacking at the moment. The assumption appears to be that Gutka consumption reduces hunger, which in turn results in an inadequate food intake and thus malnutrition including anemia. However what are the nutritional/ health implications of gutka consumption, what do we know about that from existing literature?

Construction of key variables

The study collected an impressive amount of primary data and yet a lot of that gets lost in the brief description of the research tools. I would suggest adding in a section that describes how key variables were constructed. For example, how was the 4- week food frequency score calculated and used? What are the different levels of food insecurity in the HFIAS? What was the premise behind picking seafood consumption as an indicator of a monotonous diet? Is seafood lacking in iron/ bioavailable iron? Was red meat consumption derived from the food frequency questionnaire or asked separately? How was Gutka consumption measured?

The study is impressive in its collection of biomarkers for anemia. I would like to read more about the protocol that was followed for collection, storage and transportation of blood samples. What was the refusal rate specifically for the blood draws? A brief discussion of the limitations of using hemoglobin as an indicator of anemia.

The role of the market versus own- production

There is a lot of emphasis given to the role of red meat consumption for alleviating anemia in this community. It appears that consumption is low because of a preference for other foods as well as because of a high price of meat products. There is a lot of literature that has looked at the role of markets vs own- production for improved nutritional outcomes. More recently attention has focused on the increasing unaffordability of health diets. I am sharing some references below that might be useful, particularly for South Asia:

Gupta et al (2019). Food Policy. Women’s empowerment and nutrition status: the case of iron deficiency in India. Available at https://www.sciencedirect.com/science/article/pii/S0306919219305858

FAO State of Food security and nutrition in the world (2020) - Transforming Food Systems for Affordable Healthy Diets. Available at https://www.fao.org/publications/sofi/2020/en/

Gupta et al (2021). Global Food Security. Ground truthing the cost of achieving the EAT lancet recommended diets: Evidence from rural India. Available at https://www.sciencedirect.com/science/article/pii/S2211912421000080

Policy implications

What are the policy implications of your results? What can be done to enable such communities to have access to healthy, diverse and affordable foods? Does it come from their own production, or market reforms, or alternate strategies like fortification/ supplementation possibly?

The paper will benefit from a round of copyediting.

Reviewer #2: Dear authors

Please consider the following comments for revision of your paper:

1) there are no tables in the manuscript!! I cannot comment on your results without looking at the tables.

2) I agreed to review this paper because the proposed correlation between Gutka consumption and Anemia seemed interesting, however, there is no discussion on the reason why these two should be correlated. Why should we expect these two to be correlated? this discussion is crucial for this paper and currently it is entirely missing.

3) It is customary to present the organization of the paper at the end of the introduction.

4) In sample size, it is mentioned that sample size was increased to 557 considering dropouts and non response. However, later in the paper, it is explained that total 600 women were considered and final sample of eligible women was 557. In the Sample size section, 557 should be explained better. Currently, it is confusing.

5) Data collection was done in April and May. Authors should explain if hot weather or seasonal issues can affect the results.

6) What is the nutritional value of Gutka?

7) do they consume Gutka because it is addictive or because they want to suppress hunger? If they eat their own catch and sea food is cheap, why do they need to suppress hunger?

8) Why dont they have poultry at home? it is usually a source of eggs in rural/slum households.

9) How is the food insecurity measured? How is poverty or extreme poverty measured? it is mentioned in the discussion section.

10) How did you define balanced diet and were respondents made aware of this definition?

11) In conclusion section, some recommendations for utility stores are mentioned. Since your discussion has been mostly around meat etc, do Utility Stores sell meat?

6. PLOS authors have the option to publish the peer review history of their article (what does this mean?). If published, this will include your full peer review and any attached files.

Reviewer #1: No

Reviewer #2: No

---

## [Author Response · Author response to Decision Letter 0]

26 Sep 2022

Respected reviewer, 

Greetings. Many thanks for reviewing our paper and given us very comprehensive feedback. This is very useful. We have attached a word document with all point-by-point response to the comments which were raised by editor and reviewer. 

Looking forward to hear from you. 

Regards

Ameer

---

## [Decision Letter · Decision Letter 1]

17 Oct 2022

Gutka Consumption and Dietary Partialities Explaining Anemia in Women of a Coastal Slum of Karachi, Pakistan: A Mixed-method Study

PONE-D-22-06122R1

Dear Dr. Muhammad,

We’re pleased to inform you that your manuscript has been judged scientifically suitable for publication and will be formally accepted for publication once it meets all outstanding technical requirements.

Kind regards,

Faisal Abbas, PhD

Academic Editor

PLOS ONE

Additional Editor Comments (optional):

Conditionally accept provided author incorporate comments (minor) by reviewer 2.

Reviewers' comments:

Reviewer's Responses to Questions

**Comments to the Author**

1. If the authors have adequately addressed your comments raised in a previous round of review and you feel that this manuscript is now acceptable for publication, you may indicate that here to bypass the “Comments to the Author” section, enter your conflict of interest statement in the “Confidential to Editor” section, and submit your "Accept" recommendation.

Reviewer #1: All comments have been addressed

Reviewer #2: All comments have been addressed

2. Is the manuscript technically sound, and do the data support the conclusions?

Reviewer #1: Partly

Reviewer #2: Yes

3. Has the statistical analysis been performed appropriately and rigorously? 

Reviewer #1: Yes

Reviewer #2: Yes

4. Have the authors made all data underlying the findings in their manuscript fully available?

Reviewer #1: Yes

Reviewer #2: No

5. Is the manuscript presented in an intelligible fashion and written in standard English?

Reviewer #1: Yes

Reviewer #2: Yes

6. Review Comments to the Author

Reviewer #1: (No Response)

Reviewer #2: Dear autors, thank you for your responses to my comments. I just have two minor comments before I can give it a green light:

1) your methodology for calculating food security is still unclear. Please write it as if you were explaining it to a BS student.

2) Your analysis are giving you correlations at best. Please rephrase your causal interpretations in the entire manuscript. Claiming that gutka is a major cause of anemia is a very strong statement and your analysis does not prove it.

7. PLOS authors have the option to publish the peer review history of their article (what does this mean?). If published, this will include your full peer review and any attached files.

Reviewer #1: No

Reviewer #2: No

---

## [Editor Report · Acceptance letter]

20 Oct 2022

PONE-D-22-06122R1 

Gutka Consumption and Dietary Partialities Explaining Anemia in Women of a Coastal Slum of Karachi, Pakistan: A Mixed-method Study 

Dear Dr. Muhammad:

I'm pleased to inform you that your manuscript has been deemed suitable for publication in PLOS ONE. Congratulations! Your manuscript is now with our production department. 

Kind regards, 

on behalf of

Dr. Faisal Abbas 

Academic Editor

PLOS ONE